# Optimised $CO_2$-containing medium for *in vitro* culture and transportation of mouse preimplantation embryos without $CO_2$ incubator

**Yasuyuki Kikuchi**[1], **Sayaka Wakayama**[2], **Daiyu Ito**[1], **Masatoshi Ooga**[1], **Teruhiko Wakayama**[2]*

**1** Faculty of Life and Environmental Science, University of Yamanashi, Kofu, Japan, **2** Advanced Biotechnology Center, University of Yamanashi, Kofu, Japan

* twakayama@yamanashi.ac.jp

**Data Availability Statement:** All relevant data are within the paper.

**Funding:** This work was partially funded by the Japan Society for the Promotion of Science to M.O.

## Abstract

Conventional *in vitro* culture and manipulation of mouse embryos require a $CO_2$ incubator, which not only increases the cost of performing experiments but also hampers the transport of embryos to the other laboratories. In this study, we established and tested a new $CO_2$ incubator-free embryo culture system and transported embryos using this system. Using an Anaero pouch, which is a $CO_2$ gas-generating agent, to increase the $CO_2$ partial pressure of CZB medium to 4%–5%, 2-cell embryos were cultured to the blastocyst stage in a sealed tube without a $CO_2$ incubator at 37˚C. Further, the developmental rate to blastocyst and full-term development after embryo transfer were comparable with those of usual culture method using a $CO_2$ incubator (blastocyst rate: 97% versus 95%, respectively; offspring rate: 30% versus 35%, respectively). Furthermore, using a thermal bottle, embryos were reliably cultured using this system for up to 2 days at room temperature, and live offspring were obtained from embryos transported in this simple and very low-cost manner without reducing the offspring rate (thermal bottle: 26.2% versus $CO_2$ incubator: 34.3%). This study demonstrates that $CO_2$ incubators are not essential for embryo culture and transportation and that this system provides a useful, low-cost alternative for mouse embryo culture and manipulation.

## Introduction

The current mainstream method of embryo culture *in vitro* requires a $CO_2$ incubator, which provides an optimal culture condition by stably maintaining both temperature and gas phase. However, using a $CO_2$ incubator for culture has several weak points. For instance, embryonic development in a $CO_2$ incubator stops unless $CO_2$ is continuously supplied [1]. Furthermore, in addition to the operating costs associated with a $CO_2$ incubator, there is the risk of unintentional disruption of the gas supply due to unexpected accidents or events such as power outage. Moreover, because a $CO_2$ incubator and its associated gas cylinder are relatively large, it is not

(17K15394), to D.I. (JP20J23364); the Naito Foundation to S.W.; Takahashi Industrial and Economic Research Foundation to S.W. (189); Asada Science Foundation to T.W.; Takeda Science Foundation to T.W.: and Canon Foundation for Scientific Research to T.W. (M20-0006).

**Competing interests:** The authors have declared that no competing interests exist.

suitable for embryo transport between laboratories. Embryos in culture transport had to use a dedicated unconventional mini-incubator [2]. Therefore, embryos are frozen with liquid nitrogen for transport; however, the processes of both freezing and thawing embryos are technically difficult and can negatively affect the development rate of embryos that are cryopreserved [3]. In addition, the transportation of frozen embryos using liquid nitrogen or dry ice [4] not only markedly increases the cost of transport but also mandates additional compliance to extra safety, hygiene, and regulatory transport requirements [5]. In fact, this study was initiated for the purpose of culturing mammalian embryos on the International Space Station to examine the effect of gravity on the differentiation of 2-cell embryos to the blastocyst stage. To perform this experiment conventionally, frozen embryos would have to be thawed and cultured by astronauts under microgravity condition, using a sturdy sealed device that has no gas permeability.

Previously, several methods of embryo culture without a $CO_2$ incubator have been developed. Vajta et al. placed a culture dish in a foil bag and submerged it in a thermostatic bath and successfully cultured early stage embryos without using a $CO_2$ incubator [6]. However, the quality of blastocysts was reduced. After equilibrating culture medium in advance using the conditioned air of a 5% $CO_2$ incubator, Ozawa et al. cultured porcine zygotes with the medium in a tightly closed glass tube without a $CO_2$ incubator and successfully obtained blastocysts [7]. Although the quality of blastocysts was slightly decreased, they demonstrated that embryos could be cultured without using a $CO_2$ incubator when equilibrated medium with a sealed tube was used. Similar results were also reported in a study using mouse embryos [8,9]. In those studies, the culture medium was equilibrated with a gas mixture and the embryos were cultured in a sealed microtube. Blastocysts were obtained without a reduction of the developmental rate; however, a large $CO_2$ incubator and/or heavy gas cylinder was required to prepare the equilibrated medium. To address this issue with the gas cylinder, Swain reported that the $CO_2$ can be alternatively supplied by chemical reaction via a simple handmade device provided adequate saturation of the medium with $CO_2$ [10]. In addition, the same group also demonstrated that the HEPES or MOPS-buffered medium can support mammalian embryo development outside $CO_2$ incubator without adverse effect [11]. Thus, these studies demonstrated that simpler embryo culture is possible, but still needs some improvement.

To develop a simpler embryo culture method, we previously reported an approach in which a dish placed in a plastic bag with a deoxidising agent is kept at an appropriate temperature using a thermoplate and is observable by stereomicroscope [12]. Using this system, we were able to observe embryonic development by stereomicroscopy and achieved high blastocyst and offspring rates. We also developed a small, warm box powered by a battery that could maintain an optimal temperature for embryo development. In that study, mouse embryos were transferred to a microtube, which was usually used for molecular analysis, and the tubes were transferred into a plastic bag containing a mixed gas (5% $CO_2$, 5% $O_2$). We found that this method enabled the culture of good-quality embryos and also allowed the shipment of live embryos by parcel delivery [13]. However, because this warm box is not commercially available; its adoption has not yet spread to other laboratories. Thus, although these efforts also indicate that embryos can be cultured without $CO_2$ incubator, several challenges remain, such as the low developmental rate to blastocyst and the requirement for specialised equipment. For these reasons, although a $CO_2$ incubator is more expensive than a regular incubator, $CO_2$ incubators are still commonly used for embryo culture and live embryo transportation.

The purpose of the current study was to develop a new $CO_2$ incubator-free culture method for mouse embryos. Our system consisted of culture medium containing an optimal $CO_2$ partial pressure (termed Optimised $CO_2$-containing medium or OptC medium), a tightly sealed tube to prevent gas from escaping and a heater. If this method could culture embryos, the cost

for embryo culture would be significantly decreased, with greater safety against accidents. Furthermore, this system may be a useful alternative for laboratories that do not have a $CO_2$ incubator but suddenly required to perform embryo culture experiment. We also developed a simple method for embryo transport using a thermal bottle, conducting actual transport experiments to investigate whether it is practical.

## Results

### Establish of a method for preparation an OptC medium

Plastic tubes containing CZB medium were placed in a 5% $CO_2$ incubator (Fig 1A), and the $CO_2$ partial pressure of the medium increased with longer treatment duration, achieving equilibrium of approximately 4% after 24 h (Fig 1B). During this period, the pH of the medium decreased reaching a final pH of 7.4, which is the pH for conventional culture of mouse embryo [14]. This equilibrated CZB medium was termed OptC medium, which we then further characterised and tested.

OptC medium was also produced using the gas-generating Anaero pouch instead of a $CO_2$ incubator to increase the partial pressure of $CO_2$ in the medium (Fig 1C and 1D). The rates of $CO_2$ partial pressure increase and pH decrease in the medium were more rapid using Anaero pouches compared with a $CO_2$ incubator. After 24 h of treatment, the partial pressure of $CO_2$ in the medium increased to 12% and the pH decreased to 7. Therefore, we stopped treatment at 3 h as the partial pressure of $CO_2$ in the medium was approximately 5% and the pH was approximately 7.4 (Fig 1E).

### Embryo culture using OptC medium

Using untreated medium warmed to 37°C placed in a thermostatic chamber, we found that the developmental rate of 1-cell zygotes (ICR strain) to blastocysts was significantly reduced compared with that of controls (47.9%, P < 0.001; Table 1). However, for zygotes cultured in OptC medium, we found that most embryos developed to a relatively good of blastocyst at the same rate as controls (97.4% vs. 97.9%, P = 0.771; Fig 2A–2C, Table 1). In addition, similar experiments were performed using 1-cell zygotes of the inbred mouse strain (C57BL/6N) and hybrid strain (B6D2F1), and we found that the blastocyst rates of both strains were similarly high compared with those of controls (Table 1).

When we measured the $CO_2$ partial pressure and pH of OptC medium before and after embryo culture for 4 days, the $CO_2$ partial pressure was reduced from 4.85% to 2.38%, and pH was slightly increased from 7.40 to 7.71. Similar result was obtained when medium was just kept at 37°C for 4 days without embryo ($CO_2$: from 4.86% to 2.44%; pH: 7.40 vs. 7.70). The quality of blastocysts cultured in a sealed plastic tube with OptC medium was also evaluated on the basis of immunostaining and offspring rate. For embryos cultured in untreated medium warmed to 37°C, the mean ICM and TE cell numbers were low (Total: 46.4, P < 0.001; ICM: 4.8 and TE: 41.2). However, we found that the mean ICM and TE cell numbers of embryos cultured with OptC medium were markedly higher (Total: 73.8, P = 0.11; ICM: 8.5 and TE: 65.0) and comparable with those of controls (Total: 84.1, ICM: 6.4 and TE: 77.1) (Table 2 and Fig 2D and 2E). However, when the ratio of ICM/TE were compared between all group (Table 2), blastocysts cultured with untreated and OptC medium were similar (0.14 vs. 0.13 respectively), and higher than control (0.08). Next, we evaluated embryo transfer and offspring generation. For blastocysts derived from untreated medium, we found that offspring were obtained, but the success rate was significantly low (10.5%, P < 0.001). However, for blastocysts cultured with OptC medium, many healthy offspring were obtained with a success rate similar to that found in controls (34.5% versus 43.5%, respectively, P = 0.35) (Table 3 and Fig 2F).

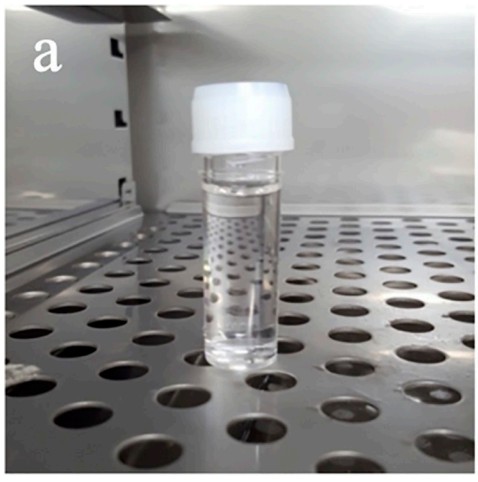
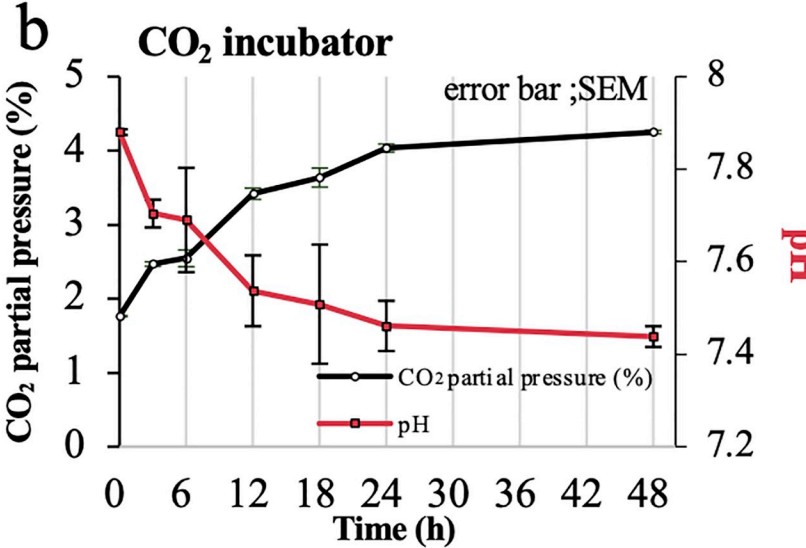
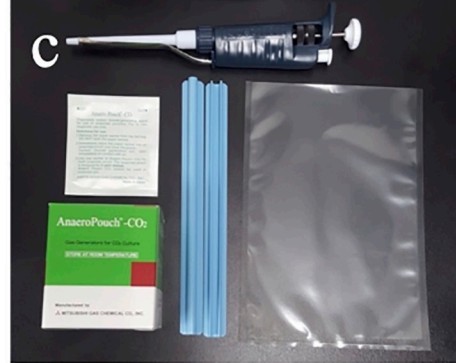
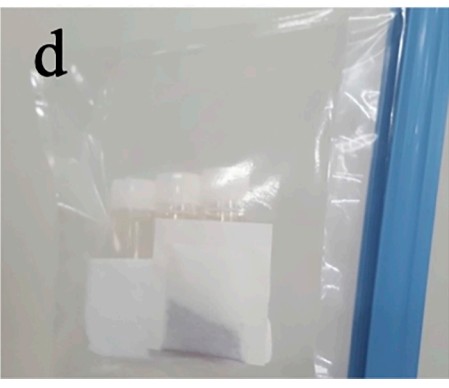
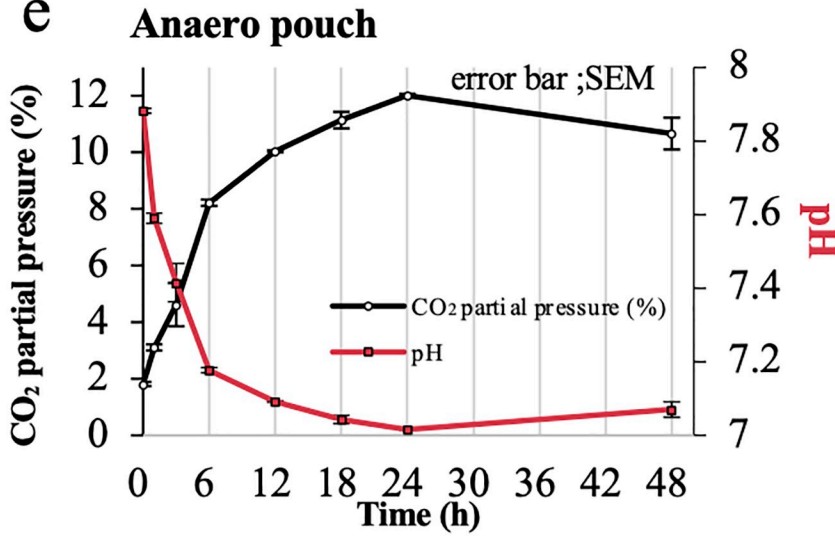

**Fig 1. Change of $CO_2$ partial pressure and pH of the CZB medium.** (a) A plastic tube containing 5 mL of CZB medium with a loose lid was placed in a $CO_2$ incubator. (b) $CO_2$ partial pressure and pH of the medium were measured up to 48 h using a $CO_2$ incubator. (c) An Anaero pouch, clips and gas barrier film. (d) An Anaero pouch with 5 mL of CZB medium in a barrier film. (e) $CO_2$ partial pressure and pH of the medium were measured up to 48 h using an Anaero pouch. Black line indicates partial pressure of $CO_2$, and the red line indicates pH, with error bar (SEM).

OptC medium allowed embryos in sealed tube to develop normally.However, this system still needed a $CO_2$ incubator for the preparation OptC medium. Therefore, we investigated whether embryos cultured without a $CO_2$ incubator at all. For this purpose,OptC medium equilibrated with $CO_2$ produced by an Anaero pouch were used for culture the embryos fertilized *in vivo* (not *via* a $CO_2$ incubator) in a sealed tube. Then, we examined whether OptC

**Table 1. Blastocyst development of embryos cultured in a sealed tube with OptC medium.**

| Mouse strain | Medium | No. embryos examined | Fragment | 1-cell | 2-cell | 4 to 8-cell | Morulae | Blastocyst |
|---|---|---|---|---|---|---|---|---|
|  | Control | 141 | 0 (0) | 0 (0) | 0 (0) | 3 (2.1) | 0 (0) | 138 (97.9) [a] |
| ICR | Untreated | 121 | 2 (1.7) | 9 (7.4) | 24 (17.0) | 7 (5.8) | 21 (17.4) | 58 (47.9) [b] |
|  | OptC medium | 151 | 0 (0) | 0 (0) | 1 (0.7) | 0 (0) | 3 (2.0) | 147 (97.4) [a] |
| B6D2F1 | Control | 49 | 0 (0) | 1 (2.0) | 0 (0) | 0 (0) | 1 (2.0) | 47 (95.9) |
|  | OptC medium | 55 | 0 (0) | 1 (1.8) | 0 (0) | 2 (3.6) | 0 (0) | 52 (94.5) |
| C57BL/6 | Control | 33 | 2 (6.7) | 0 (0) | 0 (0) | 0 (0) | 1 (3.3) | 30 (90.9) |
|  | OptC medium | 50 | 4 (8.0) | 0 (0) | 0 (0) | 0 (0) | 3 (6.0) | 43(86.0) |

[a–b] indicate a significant difference (tukey's WSD test) compared in blastocyst rate while ICR (P < 0.05). Control: Embryos were cultured on the dish covered with mineral oil. Untreated: Embryos were cultured in a plastic tube using untreated CZB medium.

medium generated by Anaero pouch instead of $CO_2$ incubator also can support embryo development to full term. In this experiment, we found that most embryos developed to blastocysts, and after transfer of those embryos into recipient females, healthy offspring were obtained at a success rate similar to OptC medium generated by $CO_2$ incubator (30.4% versus 35.2%, respectively; Table 4 and Fig 2G). Furthermore, offspring obtained from this culture method grew healthy and had normal fertility (Fig 2H).

## Culture in thermal bottle

As it was confirmed that embryo culture was possible even in a sealed tube, we next evaluated whether embryos could be cultured in a thermal bottle for mountaineering containing 38.5˚C water (Fig 3A). First, we determined how long the appropriate temperature for embryo culture could be maintained in the thermal bottle. Although the thermal bottle was placed at room temperature (~25˚C), we found that the internal water temperature decreased to approximately 33˚C after 1 day and to approximately 29˚C after 2 days (Fig 3B). Because blastocyst rate decreases to approximately 30% at a culture temperature of 35˚C while 5 days [15], maintaining the optimal temperature is inherently important. Therefore, to mimic the transport duration needed between laboratories domestically, we decided to culture embryos in a thermal bottle for up to 2 days.

In this experiment, we used 500-μL microtubes sealed with Parafilm. Cultured 1-cell stage embryos derived from IVF showed no developmental delay after 1 day of thermal bottle incubation with OptC medium, after which the embryos were collected and subsequently cultured in a $CO_2$ incubator for 4 days (Table 5). We found that most embryos developed to blastocysts (88.3%, Fig 3C) and the success rate to obtain offspring from these embryos was identical to that of controls (28.3% versus 30.5%, respectively; Table 6, Fig 3D). However, for zygotes cultured for 2 days in a thermal bottle, we found that development was delayed or stopped at the 2-cell stage (95.9%). In subsequent culture in a $CO_2$ incubator, development resumed up to the morula stage but failed to reach blastocyst (Table 5, Fig 3C), and only one offspring was obtained when those delayed 2-cell embryos were transferred (3.3%, Fig 3D, Table 6). For 2-cell embryos derived from IVF cultured with OptC medium in a thermal bottle, we found that both 1- and 2-day incubations in a thermal bottle were possible and after subsequently cultured in a $CO_2$ incubator for 3 or 2 days, most of embryo developed to blastocyst (100% and 91%, respectively, Table 5, Fig 3C).

In addition, although the offspring rate was reduced in blastocyst transfer, it was still possible to obtain a sufficient number of offspring (31.3% and 17.5%, respectively, Fig 3D, Table 6). Furthermore, when *in vivo*-derived 2-cell embryos were used, the period of incubation in a

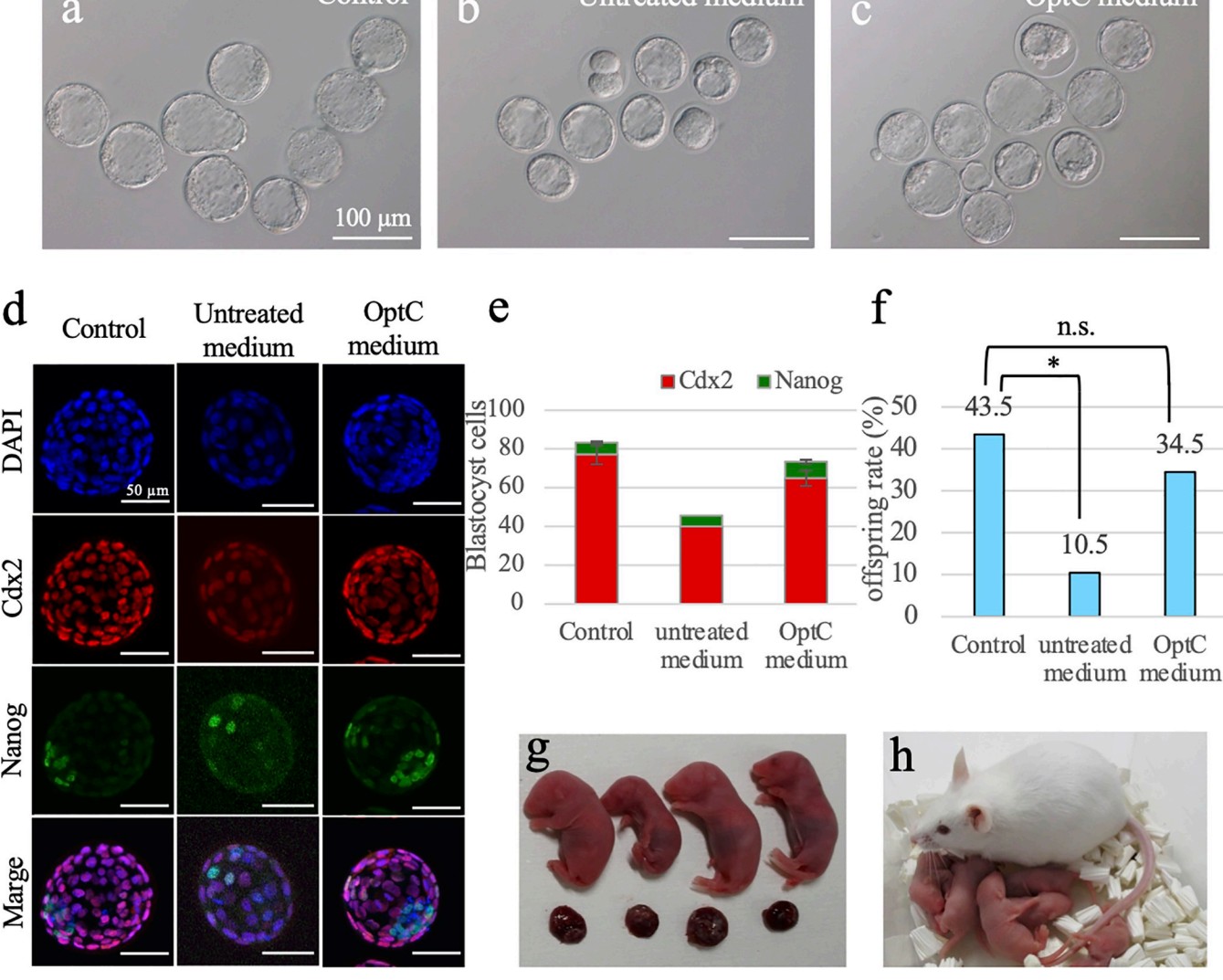

**Fig 2. Embryo culture in a sealed plastic tube using OptC medium.** (a–c) Blastocysts derived from conventional culture method (Control) (a), just warmed CZB medium in a sealed tube (untreated medium) (b), and this system (OptC medium) (c). (d) Blastocysts derived from control, untreated CZB medium with sealed tube and OptC medium with sealed tube were immunostained to measure inner cell mass (ICM) and trophectoderm (TE) cell numbers. Blue: DNA (DAPI); red: TE cell (Cdx2) and green: ICM cell (Nanog). (e) The number of Cdx2-positive (red) or Nanog-positive (green) cells. (f) The birth rate of offspring after transferred blastocysts derived from control, untreated CZB medium with sealed tube or OptC medium with sealed tube. (g) Live and healthy offspring derived from embryos cultured using OptC medium. (h) offspring obtained from OptC medium grew well and bred to the next generation.

**Table 2. Cell number of blastocysts cultured in sealed tube with OptC medium.**

| Medium | No. of embryos [*] | Total cell number (DAPI) | TE (Cdx2) | ICM (Nanog) | Ratio of ICM/TE |
|---|---|---|---|---|---|
| Control | 41 | 84.1 [a] ± 5.1 | 77.1 [a] ± 4.9 | 6.4 [b] ± 0.5 | 0.08 |
| Untreated | 19 | 46.4 [b] ± 4.0 | 41.2 [b] ± 4.0 | 4.8 [b] ± 0.7 | 0.14 |
| OptC medium | 40 | 73.8 [a] ± 4.3 | 65.0 [a] ± 4.1 | 8.5 [a] ± 0.6 | 0.13 |

[*] Embryos were randomly selected from blastocysts derived from the ICR experiment (Table 1). Control: Embryos were cultured on the dish covered with mineral oil.

Untreated: Embryos were cultured in a plastic tube using untreated CZB medium.

[a–b] indicate a significant difference (tukey's test) compared in each number of cells (P < 0.05).

**Table 3. Birth of offspring using a sealed tube with OptC medium.**

| Medium | No. embryo transferred* (recipient) | No. implantation (%) | No. offspring (%) | Mean body weight (g) | Mean placenta weight (g) |
|---|---|---|---|---|---|
| Control | 46 (5) | 26 (56.5) [a] | 20 (43.5) [a] | 1.69 | 0.14 |
| Untreated | 38 (4) | 11 (28.9) [b] | 4 (10.5) [b] | 1.99 | 0.19 |
| OptC medium | 58 (6) | 42 (72.4) [a] | 20 (34.5) [a] | 1.71 | 0.14 |

* Embryos were randomly selected from blastocysts derived from the ICR experiment (Table 1).

[a–b] indicate a significant difference (tukey's WSD test) compared in implantation rate and offspring rate (P < 0.05). Control: Embryos were cultured on the dish covered with mineral oil. Untreated: Embryos were cultured in a plastic tube using untreated CZB medium.

thermal bottle did not affect the development rate to the blastocyst (96.3% and 93.2%, Table 5). And the offspring rate was as high as that of the conventional method even when embryos were cultured for 2 days in the thermal bottle (26.2%, Fig 3D, Table 6).

We also conducted transportation experiments of embryos using a thermal bottle. A thermal bottle containing 2-cell embryos was transported on a round trip between Yamanashi and Ibaraki prefectures (a distance of approximately 200 km), and embryos were collected on the next day. Bottle transport without any additional protection may cause vibrations and changes in the ambient temperature inside the bottle; however, we found that the developmental rate to the blastocyst and offspring rate of both IVF and *in vivo* embryos was comparable with that of controls without transport (blastocyst: 96.3% and 93.3%, respectively, Table 5; offspring: 28.6% and 33.3%, respectively, Fig 3D, Table 6).

## Discussion

In this study, we successfully developed a new embryo culture system, in which 1-cell mouse embryos derived from IVF can be cultured to blastocysts *in vitro* without a $CO_2$ incubator, and furthermore, can obtain healthy offspring at a success rate similar to that of the conventional approach using a $CO_2$ incubator. Our findings clearly demonstrate that OptC medium generated by Anaero pouches has no detrimental effect on embryosand provides similar culture conditions as a $CO_2$ incubator. Using this system, mouse embryos can be simply and affordably transported to other laboratories compared with existing methods.

Conventionally, the transportation of embryos or spermatozoa requires liquid nitrogen, which makes handling difficult and transport expensive. As a solution to this problem, we previously developed a freeze-dried storage method of mouse spermatozoa [16]. Using this method, freeze-dried spermatozoa can be readily transported at room temperature even between the Earth and International Space Station [17,18]. However, a method for the freeze-drying of embryos has not yet been developed. Nakagata and colleagues reported that 2-cell mouse embryos can be preserved for several days at 4°C, and offspring were obtained from

**Table 4. Birth of offspring using a sealed tube with OptC medium generated using Anaero pouch.**

| Source of $CO_2$ to generate OptC medium | No. 2-cell examined | Frag. | 2-cell | 4 to 8-cell | Mor. | Blast. | No. embryo transferred (recipient) | No. offspring (%) |
|---|---|---|---|---|---|---|---|---|
| $CO_2$ incubator | 63 | 1 (1.6) | 2 (3.2) | 0 (0) | 0 (0) | 60 (95.2) | 54 (4) | 19 (35.2) |
| Anaero pouch | 76 | 2 (3.0) | 0 (0) | 0 (0) | 0 (0) | 74 (97.4) | 69 (5) | 21 (30.4) |

Frag.: Fragment; Mor.: Morulae; Blast.: Blastocyst.

Each rates were not a significant different ($\chi^2$ test) compared in two groups (P > 0.05).

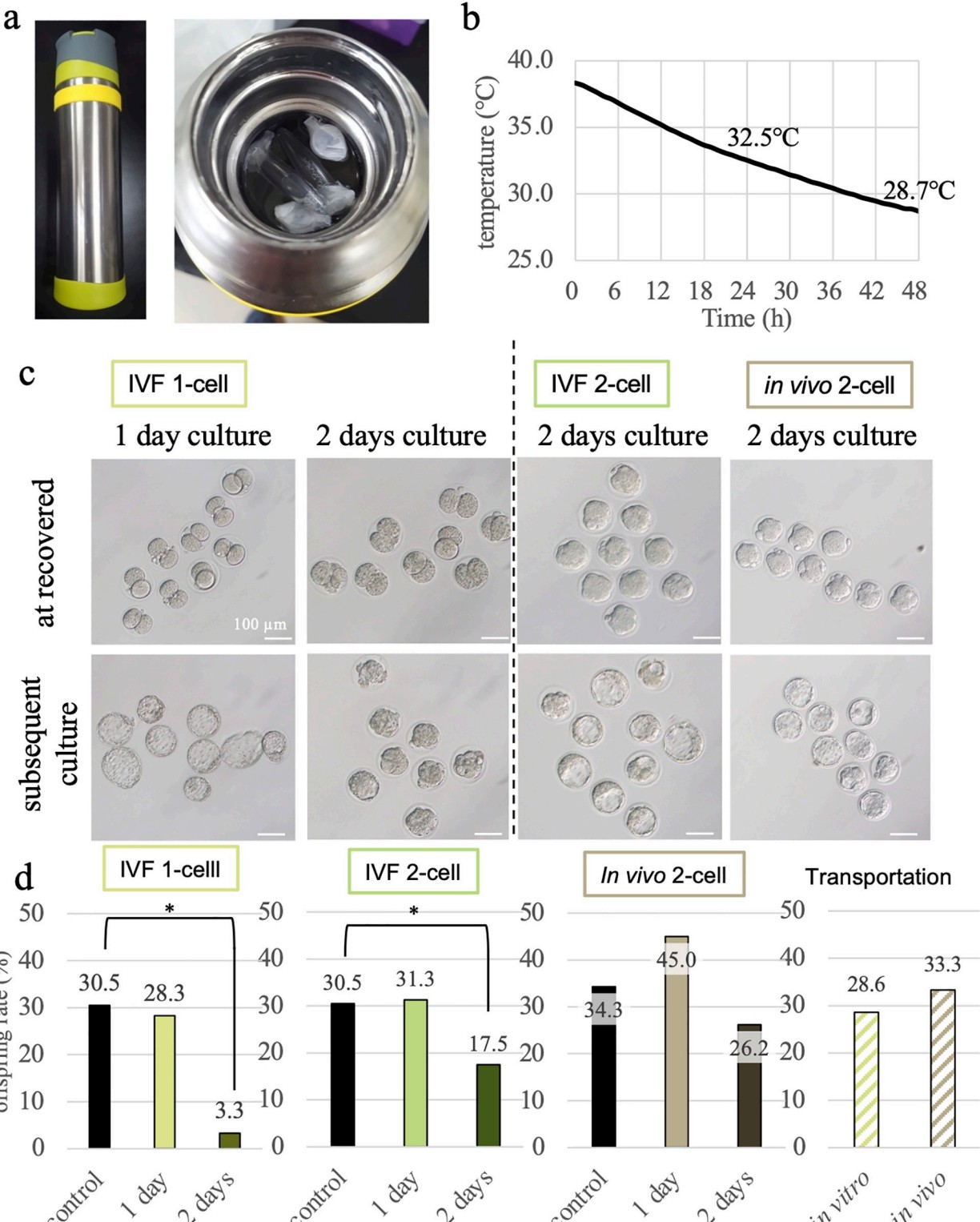

**Fig 3. Embryo culture in the thermal bottle.** (a) The thermal bottle used in this study is commercially available (left). Microtubes with OptC medium and embryos were put in the bottle containing 38.5˚C water (right). (b) Water temperature in the thermal bottle was measured for 48 h. (c) Embryos at the time of collection from the thermal bottle (upper) and embryos cultured for 3 to 4 days in a $CO_2$ incubator after collection (lower). (d) The birth rates of offspring derived from embryos cultured in the thermal bottle for 1 or 2 days.

**Table 5. Blastocyst development of embryos cultured in a thermal bottle for 1 or 2 days, followed by subsequent culture respectively, in a $CO_2$ incubator.**

| Stage of used embryos | Culture method | Culture period in thermal bottle | Subsequent culture period in a $CO_2$ incubator | No. embryo examined | Frag. | 1-cell | 2-cell | 4 to 8-cell | Mor. | Blast. |
|---|---|---|---|---|---|---|---|---|---|---|
| IVF 1-cell | Control | - | - | 111 | 1 (1.0) | 0 (0) | 2 (1.8) | 0 (0) | 2 (1.8) | 106 (95.5) [a, b] |
| | Thermal bottle | 1 day | | 60 | 0 (0) | 2 (3.3) | 58 (96.7) | 0 (0) | 0 (0) | 0 (0) |
| | | | 4 days | | 1 (1.7) | 2 (3.3) | 2 (3.3) | 0 (0) | 2 (3.3) | 53 (88.3) [b] |
| | | 2 days | | 49 | 0 (0) | 2 (4.1) | 47 (95.9) | 0 (0) | 0 (0) | 0 (0) |
| | | | 3 days | | 1 (2.0) | 2 (4.1) | 9 (18.4) | 7 (14.3) | 29 (59.2) | 1 (2.0) [c] |
| IVF 2-cell | Thermal bottle | 1 day | | 67 | 0 (0) | - | 0 (0) | 37 (55.2) | 30 (44.8) | 0 (0) |
| | | | 3 days | | 0 (0) | - | 0 (0) | 0 (0) | 0 (0) | 67 (100) [a] |
| | | 2 days | | 89 | 0 (0) | - | 1 (1.1) | 1 (1.1) | 87 (97.8) | 0 (0) |
| | | | 2 days | | 6 (6.7) | - | 1 (1.1) | 1 (1.1) | 0 (0) | 81 (91.0) [a,b] |
| *In vivo* 2-cell | Control | - | | 122 | 0 (0) | - | 0 (0) | 2 (1.6) | 2 (1.6) | 118 (96.7) |
| | Thermal bottle | 1 day | | 62 | 0 (0) | - | 0 (0) | 23 (37.1) | 39 (62.9) | 0 (0) |
| | | | 3 days | | 1 (1.6) | - | 0 (0) | 1 (1.6) | 0 (0) | 60 (96.8) |
| | | 2 days | | 44 | 0 (0) | - | 0 (0) | 0 (0) | 44 (100) | 0 (0) |
| | | | 2 days | | 1 (2.3) | - | 0 (0) | 0 (0) | 2 (4.5) | 41 (93.2) |
| IVF 1-cell | Transport by thermal bottle | 1 day | | 109 | 0 (0) | 2 (1.8) | 107 (98.2) | 0 (0) | 0 (0) | 0 (0) |
| | | | 4 days | | 1 (1.0) | 2 (1.8) | 0 (0) | 0 (0) | 1 (1.0) | 105 (96.3) [a] |
| *In vivo* 2-cell | | 1 day | | 210 | 2 (1.0) | - | 1 (0.5) | 130 (61.9) | 77 (36.7) | 0 (0) |
| | | | 3 days | | 6 (2.9) | - | 0 (0) | 5 (2.4) | 3 (1.4) | 196 (93.3) |

[a–c] indicate a significant difference (tukey's WSD test) compared in blastocyst rate while IVF ($P < 0.05$). No significant difference (tukey's WSD test) compared in blastocyst rate while *in vivo* ($P < 0.05$). Frag.: Fragment; Mor.: Morulae; Blast.: Blastocyst.

embryos transported by a courier service under refrigerated temperatures [19,20]. The developmental rate of preserved embryos decreased with increased storage time, but the approach considerably reduced transportation costs.

In contrast, embryos kept in a thermal bottle with OptC medium were maintained at room temperature for 2 days, and most of these embryos developed to blastocysts. Although the internal temperature of the thermal bottle gradually decreased, the embryos were able to slowly adapt to the lower temperature, which may have led to the higher tolerance against the lower temperature condition. The cost of transportation at room temperature is markedly cheaper than refrigerated transport by courier service, and 2 days are generally sufficient for domestic transport. Although it was shown that embryos can be adversely affected in development by physical vibration [21], the thermal bottle is filled with water, which may absorb any shocks to

**Table 6. Birth of offspring using a sealed tube with OptC medium and cultured in a thermal bottle.**

| Stage of used embryos | Culture method | Culture periods in thermal bottle | No. embryo transferred * (recipient) | No. implantation (%) | No. offspring (%) | Mean body weight (g) | Mean placenta weight (g) |
|---|---|---|---|---|---|---|---|
| IVF 1-cell | Control | - | 105 (11) | 77 (73.3) [a, b] | 32 (30.5) [a] | 1.74 | 0.16 |
| | Thermal bottle | 1 day | 53 (6) | 40 (75.5) [a] | 15 (28.3) [a] | 1.72 | 0.13 |
| | | 2 days | 30 (4) | 10 (33.3) [c] | 1 (3.3) [b] | 1.48 | 0.17 |
| IVF 2-cell | Thermal bottle | 1 day | 67 (6) | 43 (64.2) [a, b] | 21 (31.3) [a] | 1.74 | 0.12 |
| | | 2 days | 80 (8) | 44 (55.0) [b] | 14 (17.5) [a] | 1.76 | 0.26 |
| *In vivo* 2-cell | Control | - | 70 (7) | 57 (81.4) [A] | 24 (34.3) | 1.76 | 0.13 |
| | Thermal bottle | 1 day | 60 (5) | 51 (85) [A] | 27 (45.0) | 1.73 | 0.11 |
| | | 2 days | 42 (4) | 22 (52.4) [B] | 11 (26.2) | 1.83 | 0.16 |
| IVF 1-cell | Transport by thermal bottle | 1 day | 91 (7) | 75 (82.4) [a] | 26 (28.6) [a] | 1.79 | 0.15 |
| *In vivo* 2-cell | | | 99 (7) | 82 (82.8) [A] | 33 (33.3) | 1.67 | 0.11 |

* Embryos were randomly selected from blastocysts derived from thermal bottle culture testing (Table 5).

[a–c] mean significant difference (tukey's WSD test) compared in implantation rate and offspring rate while IVF (P < 0.05).

[A–B] mean significant difference (tukey's WSD test) compared in implantation rate and offspring rate while *in vivo* (P < 0.05).

the sealed tube/embryos. As we generated many offspring after transportation without a reduction in the success rate, this result indicates that a thermal bottle not only provides the appropriate temperature but also protects from physical damage, including vibrations that occur during transport.

In this study, 1-cell embryos can be cultured for only 1 day, whereas 2-cell embryos can be cultured for 2 days by thermal bottle. However, it is well known that the developmental potential of 1-cell and 2-cell embryos is affected differently by culture environment [22–25] There are additional differences between embryos derived from IVF versus *in vivo*, including mitochondrial number, freezing tolerance and even epigenetic changes [26–29]. These findings also show that tolerance against *in vitro* culture was different between 1-cell and 2-cell embryos, and between fertilisation *in vitro* and *in vivo*, both of which suggest that selection of the appropriate embryo should depend on the purpose of the experiment for *in vitro* culture.

For the study of basic biology or medicine, experimental animal such as genetically modified mice were generated in each laboratory or institute, and frequently transported between laboratories. Researchers just want to use those mice, does not want to pay effort and cost for this transportation, and therefore OptC medium and thermal bottle would be the best method because of the simplicity and less cost. The OptC medium could be made from an Anaero pouch instead of expensive $CO_2$ incubator, without reduce the success rate of offspring. In addition, OptC medium could be stored at least several days in a sealed tube at 4˚C without lose the capacity for embryo culture (unpublished observation), which has proven its potential for further application. Furthermore, this system allows us to culture mouse embryos even in an unsuitable environment, such as International Space Station by simply maintaining an appropriate temperature.

Generally, a $CO_2$ incubator is shared with several researchers at the same time, and therefore, culture conditions may become unstable because of frequent opening and closing of the door [30]. To address this limitation of conventional culture, Swain et al., suggested to use HEPES or MOPS-buffered medium, which embryos are cultured in a sealed tube and closed from the outside, allowing for stable embryo culture conditions without use a $CO_2$ incubator. Similarly, we also demonstrated that when OptC medium was used for embryos culture, most of embryos could develop to blastocyst without use a $CO_2$ incubator. Furthermore, using

OptC medium and a thermal bottle, live offspring were obtained from embryos transported at room temperature in this simple and very low-cost manner without reducing the offspring rate.

In conclusion, our culture system has applications not only for embryo culture *in vitro* and embryo transport but also basic study for understanding the mechanisms underlying mammalian embryo development without the need to purchase and maintain an expensive $CO_2$ incubator.

## Materials and methods

### Animals

Eight-to-twelve-week-old ICR female (n = 147) and male (n = 39), B6D2F1 (C57BL/6N × DBA/2) female (n = 6) and male (n = 3) and C57BL/6N mice female (n = 6) and male (n = 3) were obtained from the Shizuoka Laboratory Animal Center (Hamamatsu, Japan). The surrogate pseudo-pregnant ICR females (n = 89) that were used as recipients of the embryos were mated with vasectomised ICR males (n = 18), whose sterility was previously demonstrated. On the day of the experiment or after having finished all experiments, mice were euthanised by cervical dislocation. All experiments were conducted according to the Guide for the Care and Use of Laboratory Animals and were approved by the Institutional Committee of Laboratory Animal Experimentation of Yamanashi University with reference number: A29-24. All experiments were performed in accordance with these regulations and guidelines, which is followed in the ARRIVE guideline. All mice have been kept under SPF conditions, with controlled temperature (25C), relative humidity (50%), and photoperiod (14L-10D). They were fed a commercial diet and provided distilled water *ad libitum*. In this study, body weight was not measured except recipient female because body weight of young mice does not affect the quality of embryos.

### Preparation of OptC medium

Chatot-Ziomek-Bavister (CZB) medium [24] was dispensed into 5-mL plastic tubes (Assist tube, 60.9921.530S, Sarstedt K.K., Tokyo, Japan) that were tightly capped and stored at 4˚C until use. For the first series of experiments, tubes containing 5 mL of CZB medium were placed in a $CO_2$ incubator (APN-30DR, ASTEC CO., Ltd., Fukuoka, Japan) with the lids loose (Fig 1A) for up to 48 h. The $CO_2$ partial pressure and pH of the medium were measured at 0, 3, 6, 12, 18, 24 and 48 h. Next, for the complete $CO_2$ incubator-free experiment, tubes containing CZB medium and Anaero pouch (A-63, SUGIYAMA-GEN CO., LTD., Tokyo, Japan) were put in a gas barrier film (AP-1522, I.S.O, Kanagawa, Japan), and the film was closed using a clip (Fig 1C and 1D). Anaero pouch is a $CO_2$ generator that produce a certain concentration of $CO_2$ by consumes oxygen, which is designed to culture anaerobic bacteria and there is no cytotoxicity by this reaction. The $CO_2$ partial pressure and pH of the medium were measured at 0, 1, 3, 6, 12, 18, 24 and 48 h. For both experiments, $CO_2$ partial pressure and pH were measured using an i-STAT 1 analyser (12B1X00001000020, Abbott Laboratories, Chicago, IL, USA) with an i-STAT G3 + cartridge (12B1X00001000021, Abbott Laboratories) We termed this medium as OptC medium.

### *In vitro* fertilisation (IVF)

*In vitro* fertilization was performed as previously described [31]. Briefly, spermatozoa were collected from the cauda epididymitis of ICR, C57BL/6N or B6D2F1 male mice (>10 weeks) into 200 μL of human tubal fluid (HTF) medium [32] that was then covered with sterile mineral oil

and capacitated by incubation for 1 h at 37˚C under 5% $CO_2$ in a $CO_2$ incubator. During sperm preincubation, cumulus–oocyte complexes (COCs) were collected from the oviducts of the same strain of female mice that were induced to superovulate by consecutive injections of 7.5 IU pregnant mare serum gonadotropin (E109A, ASKA Animal Health Co., Ltd., Tokyo, Japan) and 7.5 IU human chorionic gonadotropin (hCG; E764A, ASKA Animal Health Co., Ltd.) 48 h apart. Sixteen hours after hCG injection, the mice were euthanised to collect COCs. After sperm preincubation, 5 μL aliquots of the suspension were added to droplets of HTF medium containing COCs. The final sperm concentration was approximately $1 \times 10^5$ cells/mL. Fertilised zygotes, which were determined by the extrusion of second polar body, were collected from the droplets and washed in CZB medium. The zygotes were temporarily placed in fresh droplets of CZB medium preincubated at 37˚C under 5% $CO_2$ and cultured for subsequent experiments.

### *In vivo* fertilisation and collection of 2-cell embryos

ICR females were superovulated as described and then set up with ICR fertile males after hCG injection. Plug checking was carried out early the following morning. 2-cell embryos were collected from oviducts by flushing at 1.5 days postcoitum (dpc) using HEPES-CZB [33]. Next, embryos were temporarily placed in fresh droplets of CZB medium preincubated at 37˚C under 5% $CO_2$ and cultured for subsequent experiments.

### Culture in a sealed plastic tube

1-cell stage embryos derived from IVF or 2-cell stage embryos collected from oviduct were placed in a plastic tube filled with OptC medium. The lid was then tightly closed and sealed with Parafilm. Afterwards, they were kept in a thermostatic chamber (0040534–000, TAITEC CORPORATION, Saitama, Japan) at 37˚C for 96 h for 1-cell stage embryos and 72 h for 2-cell stage embryos. As a negative control, embryos were cultured in a plastic tube and warmed by the same thermostatic chamber using untreated CZB medium instead of OptC medium. After incubation, the medium of tubes was poured in a 60-mm dish, and the embryos were collected from the dish and transferred to a preincubated dish for evaluation of embryo quality. As a control, zygotes were placed in CZB medium on a dish covered by mineral oil and cultured using a $CO_2$ incubator for 5 days.

### Embryo culture in a thermal bottle

To transport embryos, we used a thermal water bottle for mountaineering (Fig 3A, FFX-901, Thermos, Tokyo, Japan). The thermal bottle is inherently mobile and readily available for purchase and has a high heat-retaining property. In our experiments, we used a 500-μL microtube [13] as a sealed tube instead of a 5-mL plastic tube because a microtube requires a minimal volume of medium to culture embryos. The water temperature in the thermal bottle was measured every 10 min for up to 2 days using a Super Thermochron temperature logger (1922T, KN Laboratories, Osaka, Japan). Approximately 10 1-cell or 2-cell embryos derived from IVF or 2-cell embryos derived from natural mating were put into microtubes with 500 μL of OptC medium. The tubes were then placed into the thermal bottle, which was filled with water at a temperature of 38.5˚C. The thermal bottle was placed on a desk (ambient temperature approximately 25˚C) for up to 48 h. Afterwards, embryos were recovered from the tube of the thermal bottle, and either their developmental potential was evaluated *in vitro* or they were transferred into recipient females.

### Transportation of live embryos using a thermal bottle

We transferred 30–50 1-cell IVF or *in vivo* 2-cell embryos into microtubes and cultured in a thermal bottle as described above. The experiment required a high number of embryos to mimic actual transportation between laboratories. The thermal bottle containing the embryos was transported to another location without any additional transport protection. Once received, the embryos were collected, cultured for 3 to 4 days, and blastocysts were then transferred into the uteri of recipient mice.

### Embryo transfer

Embryo transfer was performed as previously described [31]. Blastocyst-stage embryos derived from embryos cultured in sealed plastic tubes were transferred into the uteri of pseudo-pregnant ICR female mice at 2.5 dpc, which were previously mated with vasectomised ICR males. On the day of embryo transfer, recipients were anaesthetised by intraperitoneal injection of the anaesthetic agents' medetomidine, midazolam and butorphanol. Between 5 and 8 embryos were transferred into each uterine horn. Offspring were obtained at 18.5 dpc via caesarean section.

### Immunostaining

To count the total number of nuclei, as well as the number of trophectoderm (TE) cells and inner cell mass (ICM) blastomeres, immunofluorescence staining of blastocysts was performed. Blastocysts were fixed in PBS containing 1% paraformaldehyde for 1 h. The fixed embryos were washed twice in PBS containing 1% (w/v) Polyvinyl alcohol (PVA-PBS) for 15 min each and then stored in PVA-PBS containing 0.1% (v/v) Triton X-100 (35501–15, Nacalai Tesque, Kyoto, Japan) overnight at 4˚C. The primary antibodies used were an anti-CDX2 mouse monoclonal antibody (1:500; BioGenex, San Ramon, CA, USA, MU392A-UC) to detect TE cells and an anti-Nanog rabbit polyclonal antibody (1:500; ab80892, Abcam, Cambridge, UK) to detect ICM cells. The secondary antibodies used were an Alexa Fluor 488-labelled goat anti-mouse IgG (1:500, A11001, Molecular Probes, Eugene, OR, USA) or Cy5-labeled goat anti-mouse IgG (1:500; ab97077, Abcam) and an Alexa Fluor 568-labelled goat anti-rabbit IgG (1:500; A11004, Molecular Probes). DNA was stained with 4′,6-diamidino-2-phenylindole (DAPI; 2 µg/mL; D1306, Molecular Probes). Total cell number was counted as DAPI positive cell.

### Statistical analysis

The number of cells were evaluated using tukey's test. the rate of embryo development, implantation and the birth of offspring were evaluated using tukey's WSD test. The Anaero pouch experiment was evaluated using $\chi^2$ test. These tests were considered to represent a statistically significant difference when P value < 0.05.

## Acknowledgments

We thank Dr. S Kishigami, Y Fujimoto and Miss. C Yamaguchi for assistance in preparing this manuscript.

## Author Contributions

**Investigation:** Yasuyuki Kikuchi, Sayaka Wakayama, Daiyu Ito, Masatoshi Ooga.

**Writing – original draft:** Yasuyuki Kikuchi, Teruhiko Wakayama.

**Writing – review & editing:** Teruhiko Wakayama.

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
