## [Decision Letter · Decision Letter 0]

20 Oct 2021

PONE-D-21-21388Optimised CO2-containing medium for in vitro culture and transportation of mouse preimplantation embryos without CO2 incubatorPLOS ONE

Dear Dr. Wakayama,

Thank you for submitting your manuscript to PLOS ONE. After careful consideration, we feel that it has merit but does not fully meet PLOS ONE’s publication criteria as it currently stands. Therefore, we invite you to submit a revised version of the manuscript that addresses the points raised during the review process.

We look forward to receiving your revised manuscript.

Kind regards,

Peter J. Hansen

Academic Editor

PLOS ONE

“We thank Drs S Kishigami, and Y Fujimoto, and Miss C Yamaguchi for assistance in preparing this manuscript. This work was partially funded by the Japan Society for the Promotion of Science to M.O. (17K15394), to D.I. (JP20J23364); the Naito Foundation to S.W.; Asada Science Foundation to T.W.; and the Takeda Science Foundation to T.W.”

“This work was partially funded by the Japan Society for the Promotion of Science to M.O. (17K15394), to D.I. (JP20J23364); the Naito Foundation to S.W.; Asada Science Foundation to T.W.; and the Takeda Science Foundation to T.W.”

Reviewers' comments:

Reviewer's Responses to Questions

**Comments to the Author**

1. Is the manuscript technically sound, and do the data support the conclusions?

Reviewer #1: Yes

Reviewer #2: Yes

2. Has the statistical analysis been performed appropriately and rigorously? 

Reviewer #1: No

Reviewer #2: Yes

3. Have the authors made all data underlying the findings in their manuscript fully available?

Reviewer #1: Yes

Reviewer #2: Yes

4. Is the manuscript presented in an intelligible fashion and written in standard English?

Reviewer #1: Yes

Reviewer #2: Yes

5. Review Comments to the Author

Reviewer #1: - Wakayama et al. present an interesting series of studies aimed at simplifying the embryo culture process by eliminating the need for a CO2 incubator and facilitating embryo transport without cryopreservaton. Since much of the need for CO2 is due to the use of bicarbonate as the buffering agent in the medium, additional discussion of other buffering agents may be useful for the Discussion. Their sealed tube system may be even easier with a medium buffered with HEPES or MOPS instead of CO2? See Swain et al., 2009, Reproductive BioMedicine Online 18(6):799-810.

Minor comments:

- Lines 66 and 71: Would "equilibrated" be better than “aerated”?

- Line 72-73: I would suggest, “…supplied by chemical reaction via a simple handmade device provided adequate saturation of the medium with CO2.”

- Lines 110-111: I would suggest, "…increase the partial pressure of CO2 in the medium.” Or “…the amount of dissolved CO2 in the medium.”

- Line 127: Please specify if these cultures were performed in sealed tubes in/on a warming device?

- Line 183: I would suggest, “…tube to develop normally.”

- Line 184: I would suggest, “Therefore we…”

- Line 186-187: I would suggest, “…using OptC medium equilibrated in a sealed tube with CO2 produced by an Anaero pouch.”

- Line 236-238: I would suggest, “…used, the period of incubation in a thermal bottle did not affect the development rate to the blastocyst.”

- Lines 297-298: Differences between 1- and 2-cell embryos in their sensitivity top culture conditions is fairly well established in mice and relevant references should be included here.

- Line 305-307: Why couldn’t this method be adapted for domestic animals?

- Line 361-362: Some additional information on how the Anaero pouches produce CO2 would be useful. Is there any other products of the reaction released into the air?

- Lines 460-462: Chi-square and t-tests should not be used for more than 2 treatments.

Reviewer #2: In this new manuscript, the authors propose a new optimised system to culture and transport mouse embryos without a CO2 incubator. This paper is a major improvement of their previous paper as all the equipment required are commercially available. The rates and quality of blastocysts and then of pups obtained are clearly very convincing. I highly recommend publishing this manuscript.

I have only minor comments:

- I hope tat in the fial version of the tables the sentences like "Control (CO2 incubator)" and " Untreated medium" will fit in one line to make it more clear, same for number and corresponding %. On the PDF I downloaded from the website, only Table 3 was easy to read.

- I would remove the last sentence of the conclusion. If ou are doing SCNT or DNA microinjections I suspect ou would need good microinjectors and CO2 incubators in your lab anyway (at least for cell culture).

6. PLOS authors have the option to publish the peer review history of their article (what does this mean?). If published, this will include your full peer review and any attached files.

Reviewer #1: No

Reviewer #2: No

---

## [Author Response · Author response to Decision Letter 0]

9 Nov 2021

Response to reviewers

We have followed the instruction of PLOS ONE's style completely. 

2. Please remove any funding-related text from the manuscript and let us know how you would like to update your Funding Statement.

We corrected acknowledgement and deleted our Funding Statement.

Review Comments to the Author

Reviewer #1: - Wakayama et al. present an interesting series of studies aimed at simplifying the embryo culture process by eliminating the need for a CO2 incubator and facilitating embryo transport without cryopreservaton. Since much of the need for CO2 is due to the use of bicarbonate as the buffering agent in the medium, additional discussion of other buffering agents may be useful for the Discussion. Their sealed tube system may be even easier with a medium buffered with HEPES or MOPS instead of CO2? See Swain et al., 2009, Reproductive BioMedicine Online 18(6):799-810.

We thank you very much for your kind and valuable comments. According to your constructive comments, we added following sentence in the text.

Line 73-76.

In addition, same group also demonstrated that the HEPES or MOPS-buffered medium can support mammalian embryo development outside CO2 incubator without adverse effect[11].

Line 343-76.

To address this limitation of conventional culture, Swan et al., suggested to use HEPES or MOPS-buffered medium, which

Minor comments

- Lines 66 and 71: Would "equilibrated" be better than “aerated”?

We thank you for the kind suggestion. We changed all the words in the manuscript. Line 66, 68, 71.

- Line 72-73: I would suggest, “…supplied by chemical reaction via a simple handmade device provided adequate saturation of the medium with CO2.”

We thank you for the kind suggestion. We changed this sentence as you suggested.

Line 73-75.

 To address this issue with the gas cylinder, Swain reported that the CO2 can be alternatively supplied by chemical reaction via a simple handmade device provided adequate saturation of the medium with CO2[11].

- Lines 110-111: I would suggest, "…increase the partial pressure of CO2 in the medium.” Or “…the amount of dissolved CO2 in the medium.”

We thank you for the kind suggestion. We have adopted the former sentence. 

Line 115.

 CO2 incubator to increase the partial pressure of CO2 in the medium.

- Line 127: Please specify if these cultures were performed in sealed tubes in/on a warming device?

We thank you for the suggestion. We described how to culture those embryos in our system. 

Line 124. 

Using untreated medium warmed to 37℃ placed in a thermostatic chamber, 

- Line 183: I would suggest, “…tube to develop normally.”

We thank you for the kind suggestion. We changed the sentence.

Line 193.

OptC medium allowed embryos in sealed tube to develop normally.　

- Line 184: I would suggest, “Therefore we…”

We thank you for the kind suggestion. We corrected it as you suggested.

Line 194.

Therefore, we

- Line 186-187: I would suggest, “…using OptC medium equilibrated in a sealed tube with CO2 produced by an Anaero pouch.”

We thank you for the suggestion. We changed sentence because it had a slightly different meaning.

Line 195-197.

For this purpose,　OptC medium equilibrated with CO2 produced by an Anaero pouch were used for culture the embryos fertilized in vivo (not via a CO2 incubator) in a sealed tube.

- Line 236-238: I would suggest, “…used, the period of incubation in a thermal bottle did not affect the development rate to the blastocyst.”

We thank you for the kind suggestion. We changed the sentence as you suggested.

Line 248¬¬–249.

Furthermore, when in vivo-derived 2-cell embryos were used, the period of incubation in a thermal bottle did not affect the development rate to the blastocyst

- Lines 297-298: Differences between 1- and 2-cell embryos in their sensitivity top culture conditions is fairly well established in mice and relevant references should be included here.

We thank you for the kind suggestion. We corrected this sentence as follow and cited following four papers.

Line 311¬¬–312.

In this study, 1-cell embryos can be cultured for only 1 day, whereas 2-cell embryos can be cultured for 2 days by thermal bottle. However, it is well known that

Line 476 Reference

We added following 3 references:

22. Lee DR, Lee JE, Yoon HS, Roh S Il, Kim MK. Compaction in preimplantation mouse embryos is regulated by a cytoplasmic regulatory factor that alters between 1- and 2-cell stages in a concentration-dependent manner. J Exp Zool. 2001;290: 61–71. doi:10.1002/jez.1036

23. Wakayama T, Maruyama Y, Imamura K, Kurohmaru M, Hayashi Y, Fukuta K. Development of early-stage embryos of the Japanese field vole, Microtus montebelli, in vivo and in vitro. J Reprod Fertil. J Reprod Fertil; 1994;101: 663–666. doi:10.1530/JRF.0.1010663

24. Chatot CL, Lewis LJ, Torres I, Ziomek CA. Development of 1-cell embryos from different strains of mice in CZB medium. Biol Reprod. 1990;42: 432–440. doi:10.1095/biolreprod42.3.432

25. Davidson A, Vermesh M, Lobo RA, Paulson RJ. Mouse embryo culture as quality control for human in vitro fertilization: the one-cell versus the two-cell model. Fertil Steril. Elsevier; 1988;49: 516–521. doi:10.1016/S0015-0282(16)59783-0

- Line 305-307: Why couldn’t this method be adapted for domestic animals?

We thank you for your kind suggestion. We delated this sentence because we also believed that our system will be useful for domestic animal too. 

- Line 361-362: Some additional information on how the Anaero pouches produce CO2 would be useful. Is there any other products of the reaction released into the air?

We thank you for the kind suggestion. We have inserted a brief sentence to explaining the specification, but the detail is not released by that company. See below. 

Line 376-378

Anaero pouch is a CO2 generator that produce a certain concentration of CO2 by consumes oxygen, which is designed to culture anaerobic bacteria and there is no cytotoxicity by this reaction.

- Lines 460-462: Chi-square and t-tests should not be used for more than 2 treatments.

We thank you for the suggestion. We re-calculated all tables using tukey’s test for the number of cells and tukey’s WSD test for the rate of embryo development, implantation and the birth of offspring.

Line 473-476

The number of cells were evaluated using tukey’s test. the rate of embryo development, implantation and the birth of offspring were evaluated using tukey’s WSD test. Both tests were considered to represent a statistically significant difference when P value < 0.05.

Reviewer #2: Reviewer #2: In this new manuscript, the authors propose a new optimised system to culture and transport mouse embryos without a CO2 incubator. This paper is a major improvement of their previous paper as all the equipment required are commercially available. The rates and quality of blastocysts and then of pups obtained are clearly very convincing. I highly recommend publishing this manuscript.

I have only minor comments:

 - I hope tat in the fial version of the tables the sentences like "Control (CO2 incubator)" and " Untreated medium" will fit in one line to make it more clear, same for number and corresponding %. On the PDF I downloaded from the website, only Table 3 was easy to read.

We thank you for the very kind comments. We modified all tables as you suggested. See Tables.

- I would remove the last sentence of the conclusion. If ou are doing SCNT or DNA microinjections I suspect ou would need good microinjectors and CO2 incubators in your lab anyway (at least for cell culture).

We thank you for the kind suggestion. We removed that sentence. The new paragraph is follow. 

Line 342-345.

In conclusion, our culture system has applications not only for embryo culture in vitro and embryo transport but also basic study for understanding the mechanisms underlying mammalian embryo development without the need to purchase and maintain an expensive CO2 incubator.

---

## [Editor Report · Decision Letter 1]

15 Nov 2021

Optimised CO2-containing medium for in vitro culture and transportation of mouse preimplantation embryos without CO2 incubator

PONE-D-21-21388R1

Dear Dr. Wakayama,

We’re pleased to inform you that your manuscript has been judged scientifically suitable for publication and will be formally accepted for publication once it meets all outstanding technical requirements.

Kind regards,

Peter J. Hansen

Academic Editor

PLOS ONE
---

## [Editor Report · Acceptance letter]

14 Dec 2021

PONE-D-21-21388R1 

Optimised CO_2_-containing medium for *in vitro* culture and transportation of mouse preimplantation embryos without CO_2_ incubator 

Dear Dr. Wakayama:

I'm pleased to inform you that your manuscript has been deemed suitable for publication in PLOS ONE. Congratulations! Your manuscript is now with our production department. 

Kind regards, 

on behalf of

Dr. Peter J. Hansen 

Academic Editor

PLOS ONE